# Species Diversity and Community Structure of Macrobenthos in the Cosmonaut Sea, East Antarctica

**Jianfeng Mou** [1,2] , **Kun Liu** [2] , **Yaqin Huang** [2] , **Junhui Lin** [2] , **Xuebao He** [2] , **Shuyi Zhang** [2] , **Dong Li** [3] , **Yongcan Zu** [4] , **Zhihua Chen** [5] , **Sujing Fu** [2] , **Heshan Lin** [2,*] and **Wenhua Liu** [1,*]

[1] Marine Biology Institute, Shantou University, Shantou 515063, China; moujianfeng@tio.org.cn
[2] Laboratory of Marine Biodiversity, Third Institute of Oceanography, Ministry of National Resource, Xiamen 361005, China; liukun@tio.org.cn (K.L.); huangyaqin@tio.org.cn (Y.H.); linjunhui@tio.org.cn (J.L.); hexuebao@tio.org.cn (X.H.); zhangshuyi@tio.org.cn (S.Z.); fusujing@tio.org.cn (S.F.)
[3] Key Laboratory of Marine Ecosystem Dynamics, Second Institute of Oceanography, Ministry of Natural Resources, Hangzhou 310012, China; lidong@sio.org.cn
[4] Center for Ocean and Climate Research, First Institute of Oceanography, Ministry of Natural Resources, Qingdao 266061, China; zuyongcan@fio.org.cn
[5] Key Laboratory of Marine Geology and Metallogeny, First Institute of Oceanography, Ministry of Natural Resources, Qingdao 266061, China; chenzia@fio.org.cn
[*] Correspondence: linheshan@tio.org.cn (H.L.); whliu@stu.edu.cn (W.L.)

**Abstract:** The Cosmonaut Sea is an under-studied area and a "white spot" for macrobenthos research. Here, we report on the species diversity and community structure of macrobenthos collected using tringle trawls on the 38th Chinese National Antarctic Research Expedition (CHINARE) in the Cosmonaut Sea, East Antarctica. A total of 11 tringle trawls were deployed at different depths across the shelf, slope and seamount of the Cosmonaut Sea. A total of 275 macrobenthic species were found from 207 to 1994 m. The species richness per station varied from 23 to 89. Echinoderms (100 species), arthropods (48 species) and mollusks (36 species) were the most dominant groups. Echinoderms and arthropods dominated in abundance at seamount stations, and echinoderms, arthropods and polychaetes dominated in abundance at slope stations, while bryozoans, corals, ascidians and sponges were abundant on the Cosmonaut Sea shelf. Depth was the major driving force influencing the distribution of macrobenthos. The main components were two core communities. One was dominated by sessile suspension feeders and associated fauna. Variants of this community include sponges and bryozoans. The other core community was dominated by mobile deposit feeders, infauna and grazers–epifauna, which included arthropods and echinoderms. The results showed that the slope (40–50° E, 65–67° S) of the Cosmonaut Sea may be an important area with complex ecological processes. The results of this study contribute to the knowledge of species diversity and communities of macrobenthos in the Cosmonaut Sea and provide monitoring data for future ecosystem health assessments and better protection.

**Keywords:** macrobenthos; diversity; community structure; Cosmonaut Sea; environmental factors





## 1. Introduction

The Cosmonaut Sea is located between 30 and 60° E in the Indian Ocean sector of the Southern Ocean [1,2], which is west of Enderby Land in East Antarctica and borders the Cooperation Sea to the east and the Lisser-Larsen Sea to the west [3]. It is an important area for fisheries and a component of the Southern Ocean ecosystem [4].

The Cosmonaut Sea is a less explored region of Antarctic waters. The first detailed surveys were carried out during the austral summer of 1972–1973 and the winter of 1973 [1,5]. The second large-scale survey was conducted during the summer of 1984 [6]. In 2006, a systematic multidisciplinary survey (BROKE-WEST) was carried out to perform a broad-scale stock assessment of marine living organisms in the Western zone of the

Indian sector of the Southern Ocean [7–9]. The results of these three surveys described the characteristics of geostrophic currents, general circulation and marine chemistry. The composition of the Protistan community, the distribution and biomass of phytoplankton, the spatial distribution and abundance of larvaceans, the structure of the mesozooplankton community and the composition and distribution of krill were described in a series of publications. To date, the biodiversity and distribution of macrobenthos in the Cosmonaut Sea are still unstudied.

Previous studies in the Antarctic region have found that the composition of macrobenthic assemblages is dominated by suspension feeders and associated fauna [10]. To date, we know that macrobenthic communities on the Weddell Sea shelf have high spatial heterogeneity in biodiversity, species composition and biomass at all spatial scales, ranging from meters to hundreds of kilometers [10]. The benthic communities in the Amundsen Sea are dominated by echinoderms [11]. The soft-bottom communities in the Ross Sea are dominated by polychaetes and bivalves [12]. However, there are fewer reports about the benthic communities in the Cosmonaut Sea.

Shallow marine benthic communities around Antarctica show high levels of endemism, gigantism, slow growth, longevity and late maturity, as well as adaptive radiations that have generated considerable biodiversity in some taxa [13,14]. The characteristics of macrobenthos render them less vulnerable to interannual variability and fluctuations in productivity [15,16]. Sahade et al. [16] highlighted glacier retreat-triggered sediment runoff, which resulted in the sudden shift from a "filter feeders–ascidian domination" to a "mixed assemblage". So Antarctic benthic ecosystems can be considered good sentinels for monitoring the effects of climate change [17]. Therefore, investigating Antarctic macrobenthos can offer valuable insights into their biodiversity and biodistribution patterns in response to environmental changes.

The main objective of this study is to describe the species composition and community structure of macrobenthos in different geographical regions and water depths using tringle trawls within the Cosmonaut Sea. This survey was conducted at the shelf, slope and seamount of the Cosmonaut Sea during the 38th Chinese National Antarctic Research Expedition (CHINARE). This is the first study to assess species composition and richness in the Cosmonaut Sea. The results establish a baseline for future research on the macrobenthos community of the region and supplement existing research on macrobenthos in the Southern Ocean.

## 2. Materials and Methods

### 2.1. Study Area

The study area was the shelf, slope and seamount of the Cosmonaut Sea (Figure 1). The survey was conducted at 11 sites at water depths ranging between 207 m and 1999 m from 28 January 2021 to 8 March 2022 (Table 1). The study area was divided into three parts: (1) Kainan Maru Seamount, including stations C2'-06 and C2'-13. (2) Slope regions (>1000 m), including stations C4-11, CA1-09, CA1-10 and C5-08. (3) Shelf regions (approximately 500 m), including stations C5'-11, C6'-08, CA3-08 and C7-11.

**Table 1.** Regional environmental variables in the study area.

| | C2'-06 | C2'-11 | C4-11 | C5-08 | C5'-11 | CA1-09 | CA1-10 | C6'-08 | CA2-09 | CA3-08 | C7-11 |
|---|---|---|---|---|---|---|---|---|---|---|---|
| Depth (m) | 1591 | 1152 | 1831 | 1924 | 445 | 1994 | 1070 | 486 | 207 | 330 | 701 |
| Temperature (°C) | 0.17 | 0.14 | −0.04 | 0.19 | −0.76 | −0.09 | 0.22 | −1.44 | −1.47 | −1.46 | 0.16 |
| Salinity (psu) | 34.67 | 34.66 | 34.66 | 34.66 | 34.43 | 34.66 | 34.66 | 34.24 | 34.20 | 34.34 | 34.64 |
| DO (mg/L) | 6.66 | 6.74 | 6.99 | 6.80 | 8.47 | 7.18 | 6.76 | 9.51 | 9.60 | 9.13 | 7.05 |
| Gravel (%) | 1.67 | 0.57 | 0.00 | 0.36 | - | 0.00 | 0.00 | 39.64 | 4.6 | - | - |
| Sand (%) | 50.06 | 55.92 | 4.78 | 6.28 | - | 7.37 | 19.41 | 49.26 | 72.06 | - | - |
| Silt (%) | 43.20 | 35.58 | 71.51 | 78.84 | - | 67.56 | 63.61 | 9.44 | 19.85 | - | - |
| Clay(%) | 5.05 | 7.92 | 23.70 | 14.51 | - | 25.07 | 16.99 | 1.65 | 3.51 | - | - |
| Mean grain size (phi) | 4.35 | 4.36 | 6.77 | 6.25 | - | 6.21 | 5.34 | 2.42 | 3.61 | - | - |
| TOC (%) | 0.11 | 0.12 | 0.61 | 0.36 | - | 0.31 | 0.67 | 0.10 | 0.08 | - | - |

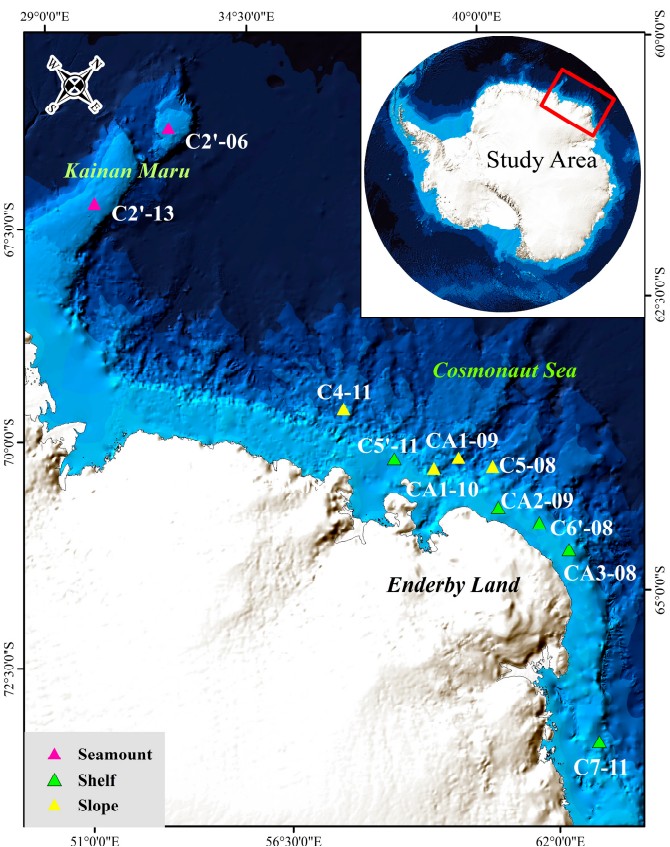

**Figure 1.** Locations of trawl stations [18].

## 2.2. Sample Processing

Triangle trawls were used to sample macrobenthos during the 38th Chinese National Antarctic Research Expedition (CHINARE) aboard the R/V *Xuelong2*. The width of each triangle trawl was 2.2 m, with a net mesh size of 20 mm; one trawl was placed at each site. It could capture specimens larger in length, which includes larger macro- and megafauna as well as some smaller animals. Once on the seafloor, the net would be trawled at approximately 2 knots with approximately 2–3 times the cable length compared to the water depth for 15 min. The trawling area, which was 2037 $m^2$, was evaluated to compensate for the qualitative nature of the triangle trawl data. The collected samples were sieved using a 0.5 mm mesh on board and then fixed in 75% ethanol. All samples were collected. A certain proportion of samples were taken by shuffling the material randomly from different parts of the entire catch because the biomass was so large at one site [19]. The numbers of individuals and masses were standardized to 1000 $m^2$ of trawled areas. The macrobenthic number and wet weight (g) were calculated per $m^2$, and the species number, density (ind./1000 $m^2$) and biomass (g/1000 $m^2$) were determined. For colonial animals, such as sponges, bryozoans and cnidarians, the counts were calculated based on one individual [16,20].

The water depth (m), temperature (°C), salinity (psu) and dissolved oxygen (DO: mg/L) of the bottom layer were measured using a conductivity–temperature–depth (CTD) meter (SBE911).

The sediments were collected using a box corer. The grain size of the sediments was determined using a laser particle size analyzer (Mastersizer 2000, Malvern Instruments, Ltd., Malvern, UK). Approximately 0.5 g of fresh sediment was pretreated with 10 mL of 30% $H_2O_2$ to remove the organic matter and then decalcified with 5 mL of diluted hydrochloric acid. The measurement error of the instrument was within 3%. The grain-size fractions were <4 μm for clay, 4–63 μm for silt, 63 μm–2 mm for sand and >2 mm for gravel.

The sediment was decalcified with 1.2 mol/L HCl, washed 6 times with deionized water and dried at 60 °C for analysis. Then, 25–30 mg carbonate-free samples were wrapped in a tin cup and analyzed for TOC using online equipment of the Elemental Analyzer—Stable Isotope Mass Spectrometer (Vario ISOPOTE Cube-Isoprime, Elementar). The instruments were first run with three blank samples, and two standard samples were also run between every 12 field samples. The analytical accuracy was ±0.02% for TOC.

No sediments were collected at stations C5'-11, CA3-08 and C7-11 because of bad weather. And the sediments were collected using a box corer.

### 2.3. Data Processing

The Shannon–Wiener diversity index ($H'_{log2}$), Margalef species richness index and Pielou evenness index were calculated for each station using species abundance. The species abundance was summed and converted to density values (1000 m$^2$). The environmental parameters of normalization variable treatments were obtained through principal component analysis (PCA) for a better understanding of the environmental gradient changes. For multivariate analysis, the species abundance per site was transformed to the fourth root to scale down the scores of the highly abundant species. A cluster analysis was performed using the Bray-Curtis dissimilarity through group average linking [21]. A similarity profile (SIMPROF) test was performed to assess different groups based on a similarity percentage to sort intergroup percentages of the average similarity contribution. A similarity percentage (SIMPER) analysis was performed to identify species that contributed to the similarity and dissimilarity of the clusters. Biota–environment matching (BIO-ENV) was used to determine the environmental factors that affect the spatial distribution of macrobenthos. All analyses were conducted using Primer 6.0 statistical software (Plymouth Marine Laboratory, Plymouth, UK) [22].

### 2.4. Classification of Macrobenthos Communities

Antarctic benthic communities have been the subject of scientific investigations since the 1950s. Gutt et al. [10] collected information from different sources and classified the macrobenthic assemblages. They are sessile suspension feeders and associated fauna (SSFA), sessile suspension feeders and associated fauna—predator-driven (SSFA-PRED), sessile suspension feeders and associated fauna—dominated by sponges (SSFA-SPO), sessile suspension feeders and associated fauna—dominated by taxa other than sponges (SSFA-OTH), mixed assemblage (MIX), very low biomass or absence of trophic guilds (VLB), "monospecific" (MONO), physically controlled (PHYCO), mobile deposit feeders, infauna and grazers (MOIN), mobile deposits feeders, infauna and grazers—infauna-dominated (MOIN-INF), mobile deposit feeders, infauna and grazers—epifauna-dominated (MOIN-EPI), vent (VENT) and seep (SEEP) [23]. In this study, we classified macrobenthos communities according to the dominant taxa.

## 3. Results

### 3.1. Environmental Parameters

The average depth of the shelf stations was 434 ± 185 m. The average depth of the deep-water (>1000 m) stations was 1594 ± 407 m. The average bottom-water temperature for all stations was −0.40 ± 0.73 °C, the maximum temperature was 0.17 °C at station C2'-06 and the minimum was −0.76 °C at station C5'-11. The average salinity of the bottom layer was 34.53 ± 0.19 psu, the maximum salinity was 34.67 psu at station C2'-06 and the minimum salinity was 34.2 psu at station CA2-09. The average DO content was 7.71 ± 1.20 mg/L, the maximum DO content was 9.6 mg/L at station CA2-09 and the minimum was 6.66 mg/L at station C2'-06 (Table 1). The average DO content was 6.86 ± 0.19 mg/L at deep-water stations and 8.75 ± 1.05 mg/L at shelf stations.

The grain size of sediments at stations C2'-06, C2'-11, C6'-08 and CA2-09 indicated sandy sediments; however, the grain size of sediment at stations C4-11, C5-08, CA1-09 and CA1-10 indicated clay sediments. The average TOC of sediments was 0.48 ± 0.18%

at stations C4-11, C5-08, CA1-09 and CA1-10, and 0.10 ± 0.02% at stations C2'-06, C2'-11, C6'-08 and CA2-09 (Table 1). The TOC of sediment in the area of 40–50° E was higher than that in the other areas in the Cosmonaut Sea.

The PCA analysis plot is shown (Figure 2). The red coordinates and red origin represent the factor scores of each site, while the black coordinates and black arrows represent the factor scores of each environmental parameter. The cumulative contribution rate of the first and second principal components is 88.6%, explaining 75.93% and 12.66% of the total variation in the dataset, respectively. Among them, PC1 has significant discrimination in sediment grain size composition and sediment organic carbon content, indicating a significant impact of sediment grain size composition on organic carbon content. Positive load represents lower sediment grain size and higher organic carbon content, while negative load represents more gravel and sand-sized particles and lower organic carbon content. PC2 has significant discrimination in water depth, DO, temperature and salinity. Positive load represents environments with shallower water depth often having higher water dissolved oxygen concentration, lower water salinity and a lower temperature, while negative load indicates that in deeper sites, DO concentration is often lower and water salinity and temperature are higher. This indicates the strength of the high-temperature and high-salinity circumpolar deep-water invasion to some extent.

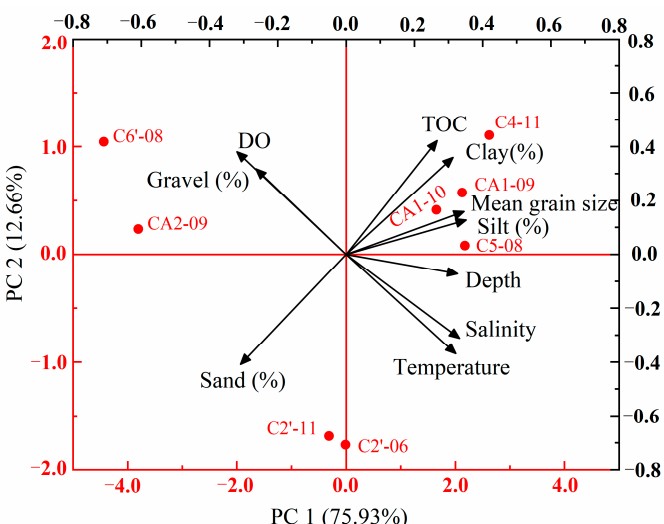

**Figure 2.** PCA plot of environmental factors at sampling stations in the Cosmonaut Sea.

### 3.2. Macrobenthos

A total of 275 macrobenthic taxa were found in the Cosmonaut Sea. There were 100 echinoderms (36%), 48 arthropods (18%), 36 mollusks (13%) and 28 cnidarians (10%) in the taxa. The remaining groups included 19 bryozoans, 15 sponges, 15 annelids, 10 chordates, 2 brachiopods and 2 hemichordates. The most frequent phyla were mollusks, echinoderms, arthropods and cnidarians, occurring at every station, followed by annelids (10 stations). The average number of taxa at the sites was 47. Taxon richness varied among stations, 23–89 per station in the study area (Figure 3). The highest taxa numbers were found at station CA2-09, followed by station CA3-08 (73 taxa) and 61 taxa at station C5'-11. The three sites with the highest richness were shelf sites, whose depths were less than 500 m.

The average abundance of macrobenthos was 343 ind./1000 m$^2$ overall and was the highest at station CA2-09 (1221 ind./1000 m$^2$), followed by 522 ind./1000 m$^2$ at station CA1-10. The average abundance of macrobenthos was higher at shelf stations (400 ± 460 ind./1000 m$^2$) than at slope stations (300 ± 178 ind./1000 m$^2$), and the average abundance of macrobenthos was 285 ± 330 ind./1000 m$^2$ at seamount stations. Echinoderms (42%) and arthropods (42%) predominated, with an average abundance of 144 ± 119 ind./1000 m$^2$. This group was followed by polychaetes (4%), mollusks (3%) and cnidarians (2%).

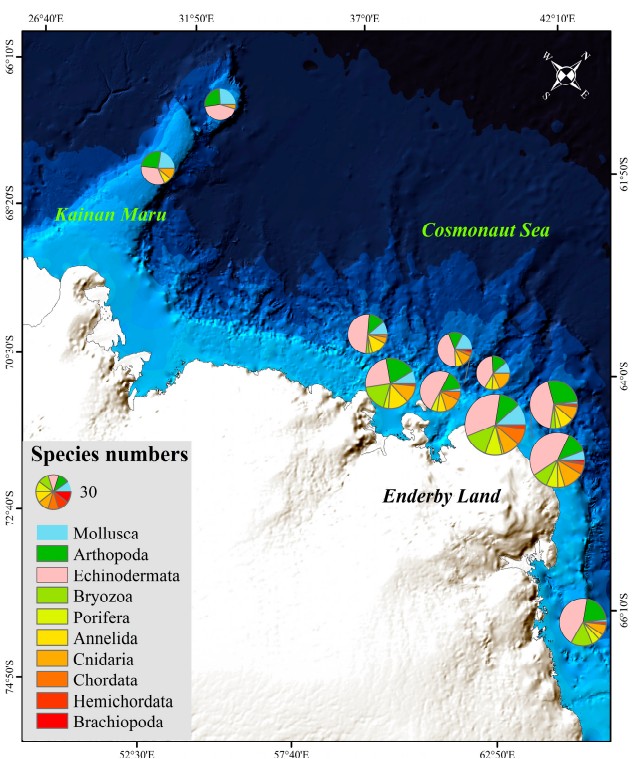

**Figure 3.** Taxon richness of macrobenthic phyla per station [18].

The average biomass of macrobenthos was 20,383.74 ± 61,701.48 g/1000 m². The average biomass was 1850.16 ± 1088.92 and 43,127.99 ± 91,273.99 g/1000 m² at the slope and shelf sites, respectively. The average biomass was 590.30 ± 185.43 g/1000 m² at seamount stations. The biomass of major macrobenthic taxa was 84% bryozoans, 5% sponges, 4% echinoderms and 2% arthropods. The biomass of bryozoans was extremely high at station CA2-09, occupying 91% of the total biomass. Echinoderms had the highest relative biomass at stations C2'-06, C2'-11, C4-11, C5'-11, CA1-09 and C6'-08. Arthropods had the highest relative biomass at stations C5-08 and CA1-10. (Table 2)

**Table 2.** Macrobenthos abundance, biomass, *d*, *J'* and *H'*(log2).

| | Phyla | C2'-06 | C2'-11 | C4-11 | C5-08 | C5'-11 | CA1-09 | CA1-10 | C6'-08 | CA2-09 | CA3-08 | C7-11 |
|---|---|---|---|---|---|---|---|---|---|---|---|---|
| | Annelida | 0 | 3 | 6 | 1 | 22 | 41 | 40 | 21 | 1 | 7 | 16 |
| | Mollusca | 10 | 3 | 12 | 2 | 4 | 2 | 1 | 0 | 52 | 38 | 0 |
| | Arthopoda | 44 | 17 | 94 | 38 | 12 | 111 | 324 | 18 | 865 | 31 | 33 |
| | Echinodermata | 464 | 27 | 211 | 46 | 114 | 105 | 119 | 196 | 101 | 83 | 117 |
| Abundance (ind./1000 m²) | Porifera | 0 | 0 | 0 | 1 | 1 | 0 | 8 | 1 | 4 | 6 | 1 |
| | Cnidaria | 0 | 2 | 1 | 2 | 5 | 1 | 24 | 5 | 30 | 10 | 2 |
| | Bryozoa | 0 | 0 | 0 | 0 | 5 | 0 | 0 | 1 | 6 | 8 | 4 |
| | Brachiopoda | 0 | 0 | 0 | 0 | 0 | 0 | 0 | 0 | 0 | 0 | 0 |
| | Hemichordata | 0 | 0 | 0 | 0 | 0 | 0 | 0 | 0 | 7 | 3 | 0 |
| | Chordata | 0 | 0 | 0 | 0 | 0 | 0 | 5 | 0 | 155 | 9 | 0 |
| | Total | 519 | 52 | 326 | 91 | 164 | 262 | 522 | 243 | 1221 | 197 | 175 |
| | Annelida | 0 | 3.97 | 29.21 | 3.10 | 29.82 | 666.04 | 122.97 | 30.56 | 8.51 | 7.58 | 62.82 |
| | Mollusca | 4.36 | 49.68 | 16.54 | 12.82 | 4.30 | 3.97 | 60.90 | 5.42 | 201.68 | 2110.95 | 0.16 |
| | Arthopoda | 40.84 | 49.54 | 706.06 | 220.38 | 10.61 | 774.64 | 1682.50 | 11.50 | 759.21 | 62.65 | 24.04 |
| | Echinodermata | 675.91 | 310.76 | 992.46 | 123.44 | 603.94 | 859.93 | 805.09 | 862.11 | 2187.06 | 548.21 | 176.61 |
| Biomass (g/1000 m²) | Porifera | 0 | 0 | 3.64 | 12.26 | 7.19 | 0 | 142.70 | 642.73 | 7699.92 | 1372.75 | 438.14 |
| | Cnidaria | 0.31 | 45.23 | 27.86 | 5.28 | 39.07 | 0.84 | 86.01 | 88.31 | 387.86 | 46.79 | 3.77 |
| | Bryozoa | 0 | 0 | 0.76 | 0 | 159.15 | 0 | 0 | 12.67 | 187,636.23 | 782.02 | 12.06 |
| | Brachiopoda | 0 | 0 | 0 | 0 | 0 | 0.19 | 0 | 0 | 0 | 0.75 | 0 |
| | Hemichordata | 0 | 0 | 0 | 0 | 49.09 | 0 | 0 | 0.60 | 3877.76 | 128.97 | 660.15 |
| | Chordata | 0 | 0 | 3.76 | 0 | 0 | 5.31 | 31.98 | 0 | 3613.54 | 272.66 | 0 |
| | Total | 721.42 | 459.19 | 1780.28 | 377.28 | 903.18 | 2310.92 | 2932.15 | 1653.90 | 206,371.78 | 5333.34 | 1377.74 |
| *d* | | 3.16 | 5.58 | 5.69 | 4.98 | 10.32 | 4.62 | 5.60 | 8.70 | 11.25 | 12.01 | 9.02 |
| *J'* | | 0.48 | 0.77 | 0.52 | 0.61 | 0.74 | 0.58 | 0.53 | 0.70 | 0.39 | 0.81 | 0.74 |
| *H'*(log2) | | 2.16 | 3.67 | 2.72 | 2.90 | 4.42 | 2.84 | 2.83 | 4.03 | 2.55 | 5.00 | 4.25 |

The average species diversity index ($H'_{\log 2}$) was 3.40 ± 0.93, the species richness index (*d*) was 7.36 ± 3.01 and the species evenness index (*J*) was 0.62 ± 0.14. The averages of the three indices for the shelf stations were 4.05 ± 0.91, 10.26 ± 1.42 and 0.68 ± 0.16, respectively. At slope stations, they were 2.77 ± 0.12, 5.20 ± 0.55 and 0.56 ± 0.04, respectively. The average values of the three indices for seamount stations were 2.92 ± 1.07, 4.37 ± 1.71 and 0.62 ± 0.21, respectively. The species diversity index, richness index and evenness index were all higher at shelf stations than in the other two areas. The species diversity index was higher at the seamount stations than at the slope stations (Table 2). The results showed that the species diversity on the shelf was higher than the slope and seamount.

### 3.3. Community Structure

The macrobenthic communities of the Cosmonaut Sea were divided into two groups (SIMPROF test, $p < 0.05$) by site type, one consisting of the shallow-water (shelf) stations and one consisting of the deep-water (slope and seamount) stations (Figure 4).

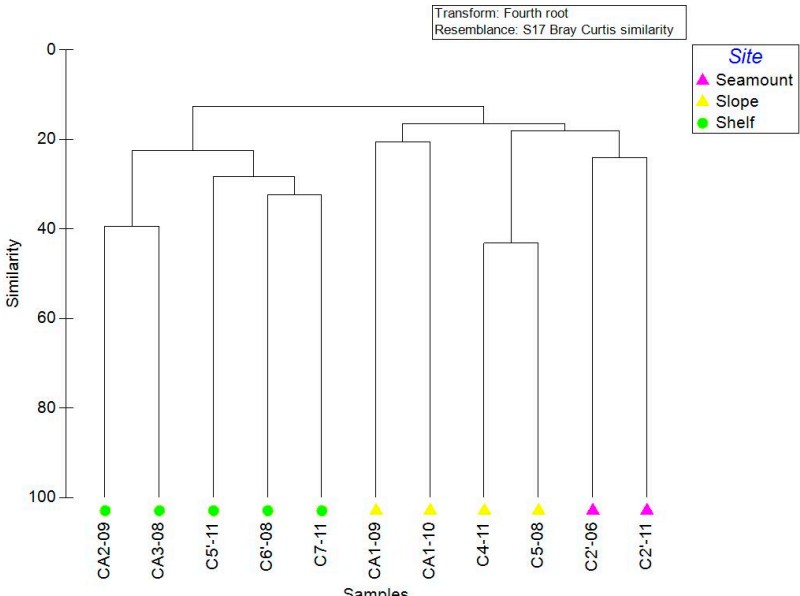

**Figure 4.** Macrobenthic community structure in the Cosmonaut Sea.

Bray–Curtis dissimilarity confirmed significant differences between the shelf stations and the deep-water stations based on the factor "depth". The SIMPER analysis showed an average similarity of 19.76% for the deep-water stations and 26.41% for the shallow-water stations, and it displayed the species most responsible for this similarity (Table 3).

In this study, the macrobenthic communities were divided into two groups. The deep-water group was mainly distributed on the slope and the Kainan Manu mountain. *Nematocarcinus lanceopes* was the dominant taxon, with a contribution of 30.03%, followed by *Ceratoserolis meridionalis* and *Ophiacantha* sp. The community type was a mobile deposit feeder community (MOIN). The community did not contain any obvious suspension feeders, while crustacean species and locally occurring ophiurids were abundant.

In the shallow-water group, the major taxa were bryozoans, sponges, cnidarians and echinoderms. The community type included sessile suspension feeders and associated fauna dominated by sponges (SSFA-SPO) and sessile suspension feeders and associated fauna dominated by others than sponges (SSFA-OTH). "Sessile suspension feeders and associated fauna" dominated by hexactinellid sponges *Rossella* sp., bryozoans including *Reteporella hippocrepis*, *Henricia* sp., *Cephalodiscus nigrescens*, *Fasciculipora ramose*, *Hornera* sp., *Orthoporidra* sp. and cold-water corals *Echinisis* sp. The associated fauna comprised the pycnogonid *Colossendeis* sp., the infauna Polynoidae and other macroorganisms that prefer an epibiotic life mode (mainly from echinoderms such as *Ophioplinthus* sp., *Amphiophiura* sp., *Ophioplinthus brevirima*, *Ophiacantha* sp,).

**Table 3.** One-way SIMPER analysis for "depth" for all stations.

| Taxa | Average Abundance | Average Similarity | Similarity/Standard Deviation | Contribution% | Cumulative% |
|---|---|---|---|---|---|
| **Group deep-water (Average similarity: 19.76)** | | | | | |
| *Nematocarcinus lanceopes* | 3.3 | 5.94 | 5.91 | 30.03 | 30.03 |
| *Ceratoserolis meridionalis* | 1.8 | 3.23 | 5.93 | 16.33 | 46.36 |
| *Ophiacantha* sp. | 1.18 | 1.58 | 0.78 | 7.99 | 54.35 |
| **Group shallow-water (Average similarity: 26.41)** | | | | | |
| *Ophioplinthus* sp. | 1.76 | 1.76 | 3.73 | 6.65 | 6.65 |
| *Echinisis* sp.2 | 1.41 | 1.3 | 4.66 | 4.91 | 11.55 |
| *Amphiophiura* sp.1 | 1.81 | 1.2 | 0.98 | 4.55 | 16.1 |
| *Reteporella hippocrepis* | 1.16 | 1.14 | 7.11 | 4.31 | 20.41 |
| *Ophioplinthus brevirima* | 1.53 | 1.01 | 1.07 | 3.82 | 24.23 |
| *Polynoidae* undefined | 1.41 | 1.01 | 1.03 | 3.82 | 28.05 |
| *Ophiacantha* sp. | 1.4 | 0.85 | 0.62 | 3.23 | 31.28 |
| *Colossendeis* sp. | 1.09 | 0.8 | 1.13 | 3.03 | 34.31 |
| *Henricia* sp. | 0.94 | 0.72 | 1.14 | 2.72 | 37.02 |
| *Cephalodiscus nigrescens* | 0.99 | 0.7 | 1.13 | 2.65 | 39.67 |
| *Rossella* sp. | 0.8 | 0.66 | 1.14 | 2.5 | 42.17 |
| *Fasciculipora ramosa* | 0.8 | 0.66 | 1.14 | 2.49 | 44.67 |
| *Hornera* sp. | 0.8 | 0.66 | 1.14 | 2.49 | 47.16 |
| *Orthoporidra* sp. | 0.8 | 0.66 | 1.14 | 2.49 | 49.65 |
| *Terebellidae* undefined | 0.99 | 0.51 | 0.59 | 1.93 | 51.59 |

### 3.4. Relationship between the Macrobenthic Community and Environmental Factors

The BIO-ENV results showed that the macrobenthos community had the highest correlation with water depth (correlation: 0.629). Then, the correlation between the combination of water depth, DO and clay was 0.615. Third, the correlation between the combination of water depth, DO, salinity and clay was 0.613 (Table 4). Water depth was found to be a very important environmental factor in macrobenthic communities in the Cosmonaut Sea, and salinity, DO and clay played relatively important roles.

**Table 4.** Biota–environment matching analysis.

| Number of Variables | Correlation | Best Variables |
|---|---|---|
| 1 | 0.629 | depth |
| 3 | 0.615 | depth, DO, clay |
| 4 | 0.613 | depth, salinity, DO, clay |
| 3 | 0.604 | depth, salinity, clay |
| 2 | 0.603 | depth, DO |
| 4 | 0.594 | depth, temperature, DO, clay |
| 2 | 0.592 | depth, clay |
| 4 | 0.592 | depth, temperature, salinity, clay |
| 5 | 0.590 | depth, temperature, salinity, DO, clay |
| 4 | 0.589 | depth, salinity, silt, clay |

## 4. Discussion

### 4.1. Taxon Richness

The overall observed taxa richness (275 taxa) in the Cosmonaut Sea was similar to that in the Amundsen Sea embayment and Pine Island (274 taxa) [11] and that in Prydz Bay (206 taxa) [16] but far lower than the overall species richness in the Southern Weddell Sea (460 taxa) [24,25] using the trawl. The taxon richness (e.g., phylum, class and order levels) in the Cosmonaut Sea was as high as that of other Antarctic regions. However, the macrobenthos collected in this study are comparatively species rich in echinoderms, arthropods, mollusks, bryozoans and cnidarians in the Cosmonaut Sea. Isopods (21 taxa),

corals (18 taxa), ophiuroids (26 taxa), asteroids (28 taxa) and holothurians (32 taxa) had higher species diversity.

The number of taxa per trawl (23–89 taxa) from the triangle trawl in the Cosmonaut Sea was higher than that reported in the Amundsen Sea (1–55 taxa) and in Prydz Bay (1–92 taxa) [26,27]. A total of 25–112 taxa were reported from trawls taken at 252–1176 m in the Southern Weddell Sea [25]. The taxa per trawl numbers in the Cosmonaut Sea were more similar to those observed in the Southern Weddell Sea at similar water depths.

*Notocrangon antarcticus* and *Nematocarcinus lanceopes* were common in the Cosmonaut Sea. Basher et al. [28] found that *N. antarcticus* was distributed on the continental shelf and upper slope, and *N. lanceopes* was distributed on the slopes, seamounts and abyssal plain in the Ross Sea. The environmental parameters that contributed most to *N. lanceopes* were depth, ice concentration, seabed slope and temperature [28]. In this study, more *N. lanceopes* were found to be distributed on the slope stations. The results of this study were consistent with those in the Ross Sea [28]. *Nematocarcinus lanceopes* was widespread throughout the Cosmonaut Sea slope.

### 4.2. Abundance

The abundance in the Cosmonaut Sea was dominated by echinoderms and arthropods. Echinoderms and arthropods dominated in abundance at the Kainan Manu Seamount, and echinoderms, arthropods and polychaetes dominated in abundance at the slope, while bryozoans, corals, ascidians and sponges stood out on the Cosmonaut Sea shelf. Griffiths et al. reported that taxon dominance varied with depth, and crustacea dominated at 200 m in abundance [20]. Holothurians at 500 m and polychaetes were most abundant at 1000 m, and crustaceans and polychaetes dominated at 1500 m. Video surveys of macrobenthos in the George V shelf revealed a dominance of bryozoans, sponges and soft corals from 117 m to 1175 m [29]. The distribution of macrobenthos on the Cosmonaut Sea shelf is similar to that on the George V shelf.

Compared to standardized macrobenthos abundances (2.5–397 ind./1000 m$^2$) from the Amundsen Sea by Agassiz trawls [11], the Cosmonaut Sea abundances (52–1221 ind./1000 m$^2$) were found to be higher at comparable depths.

### 4.3. Community Structure

The macrobenthos communities in the Cosmonaut Sea were divided based on the dominance of sessile suspension feeders, infauna and mobile deposit feeders. That can be relatively easily investigated in a quantitative or semiquantitative way. And fewer studies were conducted on general ecological processes and life history traits [30]. But the additional quantitative results can improve the classification's applicability.

Sponges, bryozoans and cold-water corals are vulnerable marine ecosystem (VME) indicator taxa. SSFA-SPO and SSFA-OTH are vulnerable communities. Anthropogenic climate change is causing the retreat of glaciers [31], and the recovery of communities is predicted to be extremely slow because growth is slow in the cold waters of the Southern Ocean. Therefore, the protection of vulnerable communities as refugia becomes more urgent if their biodiversity is to survive the accelerated rate of glacial carving [32].

### 4.4. Species Distribution and Environmental Factors

The distribution of macrobenthos at seamount stations is not clearly separated from that at slope stations but is separated from that at shelf stations. In addition, the slope in the southwest of the Cosmonaut Sea is an important area. Li et al., 2023 [33] pointed out that the upwelling of warm deep water (WDW) in the Weddell Gyre (WG) and the influx of nutrients from the Antarctic Slope Current (ASC) together lead to algal blooms, resulting in significantly higher POC concentrations in this area. It exhibits a characteristic pattern with higher concentrations to the southwest of the Cosmonaut Sea and lower values on the northeast side of the Cosmonaut Sea (Figure 5).

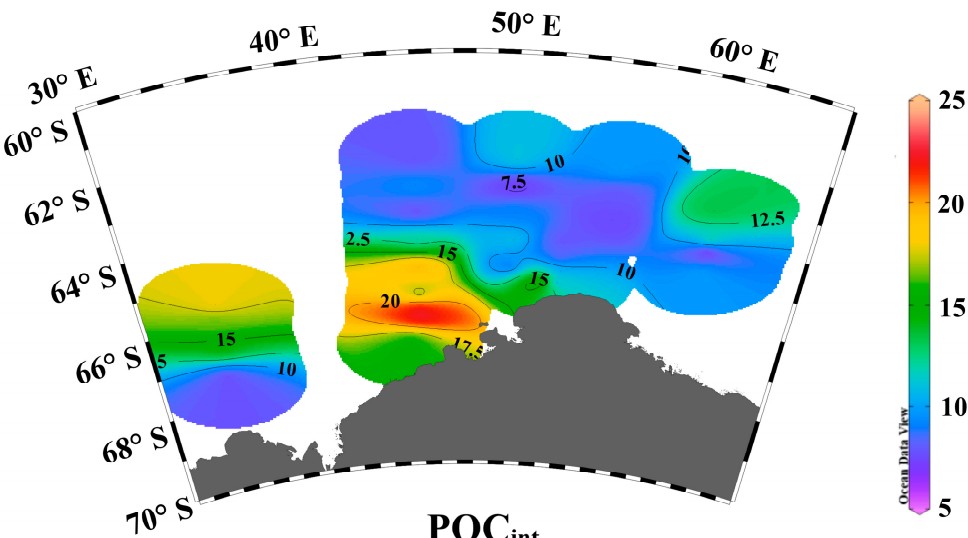

**Figure 5.** Spatial variation in integrated concentrations of particulate organic carbon (g.m$^{-2}$) in the water column (0 m–200 m) [33].

The TOC concentrations in this study were higher at slope stations, and they are food sources for polychaetes, shrimps, ophiurids and holothrians. *Dissostichus mawsoni* was observed through a photographic survey for the first time in the Cosmonaut Sea (Figure 6) [34]. Antarctic toothfish is an important fishery resource in the Southern Ocean and plays an important role in Antarctic ecosystems [15] (Barnes and Clark, 2011). West of Endby Land (40°–50° E) may be an important area in the Cosmonaut Sea. In future studies, the interspecies relationships will be analyzed to help us understand the ecological processes on the slope of the Cosmonaut Sea.

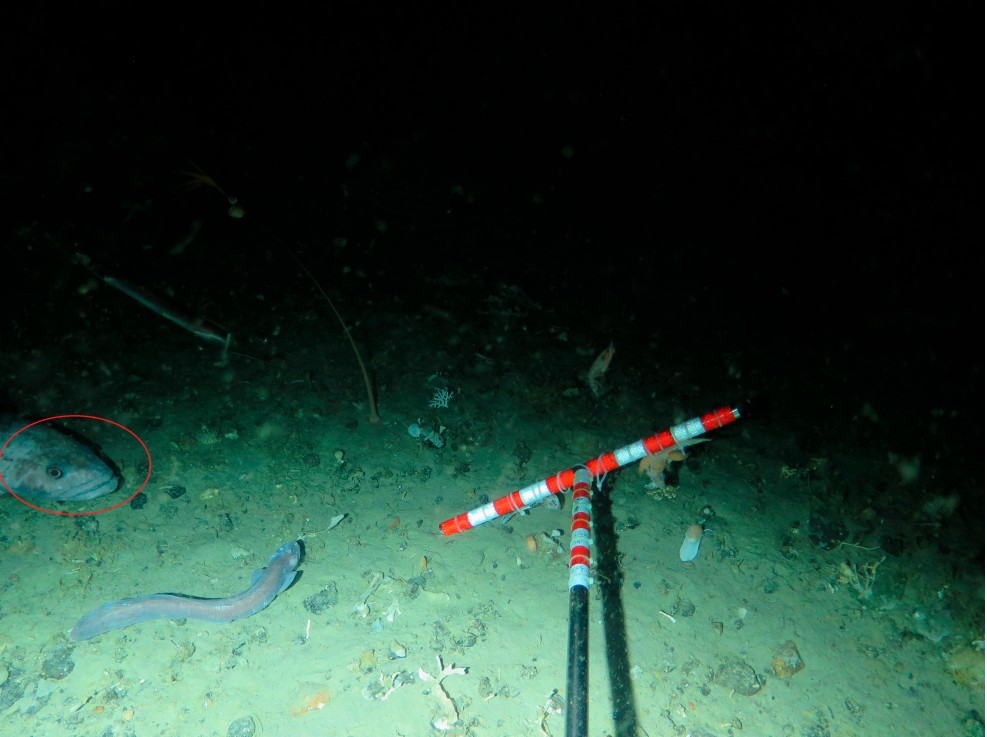

**Figure 6.** Image of *Dissostichus mawsoni* from the Cosmonaut Sea.

In this study, depth was the major driving force influencing the distribution of macrobenthos. In my opinion, the primarily vertical food supply in the upper water provided food for sessile suspension feeders, and bryozoans and sponges became the dominant taxon on the shelf. But infauna and mobile epifauna were controlled by vertical phytodetritus flux and soft sediments at the slope.

## 5. Conclusions

In this study, we found 275 macrobenthic taxa in the Cosmonaut Sea. Echinoderms were the dominant taxon, accounting for 36% of all taxa, followed by arthropods (18%). The number of taxa and the species diversity index were highest at the shelf stations. The macrobenthos community structure differed between the shelf stations and the slope/seamount stations. Water depth was the primary factor influencing community composition. The studies have revealed the distribution of macrobenthic organisms at various water depths and distinct geographic regions. The slope station area is a unique region where the POC concentration is significantly higher. Additionally, the abundance of *Nematocarcinus lanceopes* was very high, while *Dissostichus mawsoni* was observed in this area. This study has some limitations, such as the limited number of stations, which do not fully reflect the distribution of macrobenthos in the Cosmonaut Sea. Future studies can further expand the sampling range and increase the number of stations to more comprehensively reveal the distribution characteristics of macrobenthos. In the future, the urgent need for the prediction of ecosystem responses to large-scale environmental changes in Antarctica calls for the continuation of surveys in the community. This is important for maintaining the stability of the ecosystem.

**Author Contributions:** J.M.: investigation, data curation, writing—original draft preparation and validation. K.L.: conceptualization, software and writing—reviewing and editing. Y.H.: data curation and software. J.L.: data curation and writing—reviewing and editing. X.H.: data curation and writing—reviewing and editing. S.Z.: data curation. D.L.: data curation and writing—reviewing and editing. Y.Z.: investigation and writing—reviewing and editing. Z.C.: investigation. S.F.: investigation and supervision. H.L.: supervision and conceptualization. W.L.: supervision, conceptualization and resources. All authors have read and agreed to the published version of the manuscript.

**Funding:** This work was supported by the National Natural Science Foundation of China under contract (No. 42076237, No. 42106227, No. 42006127, No. 42206252), the Impact and Response of Antarctic Seas to Climate Change (IRASCC2020-2022-NO.01-02-02B & 02-03) and the National Key Research and Development Program of China (2019YFA0607003).

**Institutional Review Board Statement:** Not applicable.

**Informed Consent Statement:** Not applicable.

**Data Availability Statement:** All data presented in this study are available.

**Conflicts of Interest:** The authors declare no conflict of interest.

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
