# Peer review of "Species Diversity and Community Structure of Macrobenthos in the Cosmonaut Sea, East Antarctica"

_diversity, doi:10.3390/d15121197_

Round 1
Reviewer 1 Report
Comments and Suggestions for Authors
Diversity-2718698
Comments and Suggestions for Authors
This MS analyzed species diversity and community structures of macrobenthos in shelf, slope, and seamount of the Cosmonaut Sea. This information is considered to provide valuable information for the study to seasmount ecology. The reviewer suggests that the following few points are needed to revise for publication in Diversity.
Line 93: When the triangle trawl sampling, how can you measure the environmental data? The site of CTD data is one point, but the site of trawl is area. So, it is need the hypothesis that the area of trawl is homogeneous. Please explain the hypothesis
Line 97-98: Please check the area: it is needed to be same units between density and biomass.
density: /1,000m2 and biomass: /m2
Line 105: Please explain 1) how can the sediments sampled using triangle trawl (20mm mesh size). 2) Also, some sites in Table 1 why the sediment data did no show.
Line 122: SIMPROF test is needed to determine the differences among communities. So, ANOSIM test is need to change to SIMPROF test.
Line 131: In all environmental data, add the standard deviation
Line 178: Please change the area to 1000m2
Line 191-192: Please add the statistical analysis to explain the difference
Line 199-202: I could not understand the Table 2. If this table is the result of SIMPER, it is need to change the contribution species at each site from the result of dissimilarity.
Line 211: Please show the results of SIMPROF test.
Line 212: What it the group Deep-water or Group Shelf in Figure 6?
When you explain the difference of group after clustering, the SIMPROF test is needed.
Line 225: Some sites did not have the sediment data, how can you analysis the BIO-ENV test ? Please explain the materials and methods the methods

Reviewer 2 Report
Comments and Suggestions for Authors
General comment:
The manuscript of Mou et al. titled ‘Species Diversity and Community Structure of Macrobenthos in the Cosmonaut Sea, East Antarctica’ provides (or at least, tries) a short and incomplete description of the benthic fauna inhabiting the Cosmonaut Sea. Incomplete, since little to no information on species is provided, neither in the main body of the article nor in the supplementary material of the manuscript. The alternative would be to cite a data set deposited in a data repository, but this is also not the case. Furthermore, the structure of the community or its relation to environmental parameters is poorly described. In this context, simply enumerating results as if reading a table is not how we should describe the community structure. The text can be compact, yes, but should properly explain the main characteristics of the community or sub-communities found in the study area, which is not the case in this manuscript. Well, in the discussion there is a short description in which the communities are classified following the scheme used by Gutt (2007) and Turner et al. (2009), but that should have been the way of describing them in the results. All things considered, if the aim is ‘to supplement existing research on macrobenthos in the Southern Ocean’, they do fill the aim, but in a poor way.
The methods are relatively well written. The biological part needs some clarification on which dis-/similarity metrics were used for each analysis, whereas the environmental part needs clarification on sampling methodology and classification (sediment), data treatment for the PCA, and how potential co-correlation of environmental factors was dealt with before applying the BIO-ENV analysis.
The results, as stated above, have no fluency, and seem as if the results from a table were written down as the author read them. There are also considerations needed when conducting grouping which are not considered: E.g., instead of using a cluster analysis (+SIMPROF) to define station groups, or use the groups of the environmental PCA, they arbitrarily group stations into two groups (which are not described at all!). While not inherently wrong, it ends up on biases, and also a justification as to why they chose to do this is missing. The tables and figures need more clarity (especially the figures!) and a more explanatory legend.
The discussion lacks depth, they describe similarities and differences with other Antarctic regions, but not discuss possible explanations for it, nor details on where the similarities lay in terms of species / taxa / community types. Same applies for the role of environmental variables in structuring the benthic structure found at each station or station group/community.
As it is, I must recommend the rejection of the article, as I see key flaws in how it was made. I would recommend the authors to take example of some of the literature they didn’t considered such as Gerdes et al. (1992, Polar Biology), Gutt & Starmans (1998, Polar Biology), Grange & Smith (2013), Pineda-Metz et al. (2019) or Cummings et al. (2021, Frontiers), to get an idea on how such a descriptive work should look like, how the community needs to be described, and how to discuss the role of environmental factors in the structure and distribution of benthos.
Specific comments:
Figure 1: The text of the axis and labels needs to be larger and easier to read, the chosen color combination does not help to read the station numbers. There should be a legend to the meaning of the colors. Please mention where you got the data for the map, in this case, it seems to be the IBCSO chart (not sure if the first (Arndt et al. 2013) or second (Borschel et al. 2022) version of it), so please include the citation from it.
Line 93-96: Yes and no, you can extrapolate to an area, but the best you’ll get from a net is a semi-quantitative data set, but no ‘true’ quantitative data set.
Line 96-97: Did you sort out, count and weight all the material in the net? Or did you follow a subsampling protocol in which you took a portion of the catch, and then extrapolated for the whole catch?
Line 105-113: How did you take your sediment sample? This is not specified.
Line 115-116: Have you considered using biomass instead of abundance, so that you can better represent colonial and large organisms which tend to be ecosystem engineers?
Line 116-117: You converted species abundance to abundance? This makes little sense.
Line 120-123: Did you also use the Bray-Curtis similarity matrix for the SIMPER and ANOSIM?
Line 123-124: Then you not only did a similarity matrix but two, correct? Why did you use Euclidean distance instead of Bray-Curtis dis-/similarity? Why did you use a PCO and not an MDS?
Line 127: What is it that you mean by ‘morphologic treatment’? Did you transform your environmental data or not? Which distance (dis-/similarity) metric did you use for the PCA?
Line 139: According to Table 1, the minimum DO content was 6.66 mg/L at station C2’-06.
Line 141-143: No classification scheme was provided for sediments, it would be good if you mention in the methods section, how you classified your sediments based on type.
Line 150-151: I thought, depth was already mentioned at the beginning of the section, why repeat this?
Line 151-152: Again, new information on sediment content, which is not represented in Table 1, nor mentioned in the methods.
Figure 2: As for Fig. 1, the text (plot in general) is unreadable, the quality of the image is poor. The variance explained by the PCs is missing. The explanation of which PC explained more, how much they explain in total, and which variable is most related to each should also be provided in the figure’s legend in a compact way.
Line 160: The word ‘species’ or ‘taxa’ is missing at the end of the sentence.
Line 160-162: How is it that you have annelids before richer groups such as cnidarians and bryozoans?
Line 162-163: This is a repetition of the previous sentences and can be thus removed.
Line 163: What do you mean when you say ‘common’?
Line 166: This is not clear at all in Fig. 3, for this, you should need an extra row in Table 1, or add a table showing the biological parameters at your stations, i.e. H’, richness, evenness, S, etc. Optimally, you’d have either your species list available in a data repository, or as supplemental material, to show the details of the classification scheme you used (see e.g., Gutt and Starmans 1998, Polar Biology).
Line 166-168: Shouldn’t this information be all together in one sentence? Also, be consistent on how you present your information, if you start presenting all groups and stations with their corresponding number of taxa or richness, then do so throughout the text.
Figure 3: Again, there’s a need to improve the quality of the image, and provide a better description in the legend: What do the size of the points entails? Which station is which? Where did the chart originate (see comment for Fig. 1)? Why do you use the same color gradient for unrelated groups? E.g., both arthropods and bryozoans are in green? The colors should also help to discern between groups. The taxa should: 1) Be ordered in a clear taxonomic order and not in a random (as now) nor alphabetical order, and; 2) Either use a common name (e.g., brachiopod, or as in the text, annelids), or the Latin name, but not both.
Line 175-176: 42% each or in total? Same for the average abundance, do you mean each or only one of them, or both?
Figure 4: All comments made on style for Fig. 3 apply also to this one. You should consider removing the percentages which overlap with the bars, some of them are unreadable due to the color pallet (not to mention the low quality of the image).
Figure 5: You can spare this sort of graph which are better presented as a table, combined with the abundances and biomass per station.
Line 197-199: What was the point of doing the cluster, if you already divide them arbitrarily? Not to mention, what was the point of always mentioning three groups? If you look at the cluster, you could argue that you have 3 or even 4 groups. It would be prudent to think of using a cluster+SIMPROF approach, to clearly sort stations into groups, rather than arbitrarily define groups.
Table 2: You need to better explain what is what in the table. If you want to use short versions of ‘Average Dissimilarity’, ‘Contribution to dissimilarity in %’ and ‘Cumulative Contribution to dissimilarity in %’ go for it, but explain this in the legends, not everyone knows or can guess (as I did) what each thin means. Also, clearly show which groups you’re comparing, the average dissimilarity between groups, but also within groups is important to be mentioned, as it will greatly influence the results of your ANOSIM (hence my previous recommendation of grouping based on cluster+SIMPROF).
Line 205: I had to get to results to know which similarity matrix was used for the ANOSIM.
Line 206: What is 0.0013? the number alone is not enough.
Line 207-208: How is it that we come back to SIMPER results, and these are not included in the previous paragraph? Consider that the within group similarities are extremely low! This, at least for the ANOSIM, greatly influences and biases the result.
Table 3: The lack of needed explanation which was noted for table 2 are also here. Furthermore, as already mentioned, this should be in a single table, i.e. merge table 2 and 3 together.
Figure 4: As mentioned before, re-think how you want to use the cluster. If you already are sorting your stations, and the only thing you want to show is similarity between station groups, then there’s no need for the cluster. However, the figure shows that you have more than 2 groups, even maybe up to 5 groups (based on cluster and PCO, 4), but this is not clearly tackled in the manuscript. Finally, this particular figure, does NOT show community structure.
Line 213-214: I partly disagree. While it’s true that the depth gradient applies along the x-axis, there’s another factor regulating the spread along the y-axis, the question you need to tackle is, which factor? This is important, since the variance each axis explain is quite similar.
Table 4: Is it really necessary to have a table (in the main body of the manuscript), from which you only use/need the first three rows, and can be only included as part of the text?
Line 228: If you have families included in your ‘species’ list, the you’re talking about taxa rather than species; unless you have morphotypes or OTUs for your families.
Line 242: Reference is missing.
Line 244: Reference for Basher et al. is missing.
Line 249: Previous studies such as? And conducted where?
Line 250: Such as?
Line 256: Reference for Griffiths et al. is missing.
Line 261: Reference for Barry et al. is missing.
Line 268: Again, this needs to be better explored with the analysis and arbitrarily classification used.
Line 268-278: This is rather results than discussion, and it’s a lot better described than what was done in the results section…
Line 278-284: Conservation is nice and all, but how will this communities be affected by the mentioned environmental changes that will take place in the Cosmonaut Sea?
Line 292: Which area is ‘this area’.
Line 292-294: You need to be clearer in what you mean, you’re jumping from the Weddell Sea to the Cosmonaut Sea without any clear comparison or connection.
Figure 8: Why is there a results figure here?
Figure 9: Another Figure that corresponds to either results or supplementary material. You need to clearly point to the Dissostichus mawsoni in the picture, as not everyone is familiar with how this fish looks.
Section 4.4.: There’s all but a proper discussion on how environmental factors and distribution of the benthic taxa you found
Reviewer 3 Report
Comments and Suggestions for Authors
General Comments. The manuscript aimed to evaluate the biodiversity of the benthos in the Cosmonaut Sea. The specific objectives are very standard for this kind of study, but the novelty resides in the study area, an under evaluated region of the Antarctic Region. The manuscript certainly has a lot of effort in the sampling department, although the lack of replicates certainly compromises a more robust evaluation of the results. Despite its potential, I believe the manuscript fall short in several areas. The results were hard to follow because the coding of the stations were not meaningful and the data analysis and presentation were confusing. I do not agree with all methods applied to analyze the data, but I reckon that they are all valid approaches (although alternatives could be considered). However, the discussion really suffers from the generic approach of the objectives. Most of the discussion is dedicated to comparisons with other studies, even if robustness of such comparisons may be flimsy due to different sampling efforts and methods applied among studies. Nearly all figures require some kind of adjustment, as they were hard to understand or read most of the time (although this could be a problem during submission). The discussion also lacks appropriate referencing to other studies in a lot of statements where they are required. It also introduces a discussion regarding functional characteristics that are not presented at all in the results. Considering the novelty potential of the dataset and its interest to the audience of the journal, I believe the manuscript can be returned to the authors for a major review. If the authors believe the issues raised in the review can be addressed, then the manuscript can be reconsidered. I have attached a pdf. file with multiple comments. Please, be aware that I do not expect authors to agree with all my suggestions, but I ask to provide a suitable justification, so I can understand the motivations.

Comments on the Quality of English LanguageThe manuscript requires an extensive review of language. I am not a native speaker, so I did not suggested all corrections needed. ALthough I was able to understand the manuscript, it was hard to follow at some instances, especially during the results and discussion.
Round 2
Reviewer 1 Report
Comments and Suggestions for Authors
Diversity-2718698
Comments and Suggestions for Authors
This MS analyzed species diversity and community structures of macrobenthos in shelf, slope, and seamount of the Cosmonaut Sea. This information is considered to provide valuable information for the study to seasmount ecology. The revised MS is corrected faithfully compared to the first version. So I recommend that it be published as is.
Author Response
Thank you for your comments
Reviewer 2 Report
Comments and Suggestions for Authors
Albeit the manuscript has improved, there are still some comments that need to be addressed:
Fig. 1: Why did you included this map? This is already shown in the figure in which your stations are plotted.
Line 68-69: He highlighted this for the Antarctic Peninsula, under the special conditions that Potter Cove has. You need to clarify that this is an extremely local example, as for other fjords in the West Antarctic Peninsula, no such thing was observed.
Fig. 2: Please mention where you got the data for the map, in this case, it seems to be the IBCSO chart (not sure if the first (Arndt et al. 2013) or second (Borschel et al. 2022) version of it), so please include the citation from it.
Line 101-102: Since you used a subsample, you need to mention this in the results, as e.g. Gutt et al. (2016, https://doi.org/10.1007/s00300-015-1797-6) mentioned in their manuscript.
Line 110-115: How sediment samples were taken is still not specified in the methods section. It should be included there.
Reply to ‘I have tried using biomass and abundance, and found the results of using abundance were better, can reflect the traits of different areas’: Of which traits are you talking about? The original manuscript had no true traits analyzed separately, only taxa. If you were to transform your abundance per taxa to abundance per trait, you’d likely have an underrepresentation of the sessile suspension feeders, which is represented by the sponges and bryozoans. Biomass has been suggested as better suited (e.g. Gutt et al. 2016) as it plays a higher role in the response of benthic systems, e.g. to climate change (e.g. Jones et al. 2014 https://doi.org/10.1111/gcb.12480, Barnes et al. 2019 https://doi.org/10.1111/gbc.15055, Pineda-Metz et al. 2023 https://doi.org/10.1038/s41467-020-16093-z).
Line 131-140: The two questions I had for this section are still unanswered:
- Then you not only did a similarity matrix but two, i.e. one for the SIMPER and Cluster+SIMPROF, and a second one for the PCO and BIO-ENV? This needs to be clarified in the text.
- Why did you use Euclidean distance instead of Bray-Curtis dis-/similarity for the PCO?
Line 132-135: The point of the SIMPROF isn’t to identify characteristic species, but to determine significant clusters or groups. For identifying characteristics species you’d have to do an Indicative Species Analysis (ISA) or similar.
Line 141: Constructure? What you’re doing is classification of the community, the description of the types was already done in the cited literature.
Line 142-152: As it stands, this is rather introduction than materials and methods. Here, you should show what was done with this classification. Not the story behind it.
Line 190-191: Why are chordates listed as the last, when they had more taxa tan brachiopods and hemichordates?
Line 228: It should be ‘SIMPROF’
Fig. 5: As the figure stands, is not clear where the cut was made. You can change the options of the plot in PRIMER to separate the groups (if I recall correctly). The other alternative, is to play with the symbols and colors, as to show which stations are in the seamounts, slope, shelf, and which are your two groups.
